# Patient satisfaction with advanced physiotherapy practice internationally: Protocol for a systematic mixed studies review

**Chris Davis** [1,2]*, **Tim Noblet**[1,3], **Jai Mistry**[1,3], **Katie Kowalski** [1], **Alison Rushton** [1]

**1** School of Physical Therapy, Faculty of Health Sciences, Western University, London, Ontario, Canada, **2** Nuffield Health Learning Foundation, Nuffield Health, Surrey, England, **3** Therapies Department, St Georges University Hospitals NHS Foundation Trust, London, England

* cdavi2@uwo.ca

**Data Availability Statement:** No datasets were generated or analysed during the current study. All relevant data from this study will be made available upon study completion.

## Abstract

### Rationale

Patient satisfaction is a complex construct consisting of human and system attributes. Patient satisfaction can afford insight into patient experience, itself a key component of evaluating healthcare quality. Internationally, advanced physiotherapy practice (APP) extends across clinical fields and is characterised as a higher level of practice with a high degree of autonomy and complex decision making. Patient satisfaction with APP appears positive. While evidence synthesis of patient satisfaction with APP exists, no systematic review has synthesised evidence across clinical fields. Therefore, the objectives of this systematic review are 1) to evaluate patient satisfaction with APP internationally, and 2) to evaluate human and system attributes of patient satisfaction with APP.

### Materials and methods

A systematic mixed studies review using a parallel-results convergent synthesis design will be conducted. Searches of Medline, Embase, Web of Science, CINAHL, Cochrane, PEDro and grey literature databases will be conducted from inception to 18/7/2023. Studies of APP (World Physiotherapy definition) whereby practitioners a) have advanced clinical and analytical skills that influence service improvement and provide clinical leadership, b) have post-registration masters level specialisation (or equivalence), c) deliver safe, competent care to patients with complex needs and d) may use particular occupational titles; that measure patient satisfaction across all clinical fields and countries will be included. Two reviewers will screen studies, extract data, assess methodological quality of included studies (mixed methods appraisal tool), and contribute to data synthesis. Quantitative data will undergo narrative synthesis (textual descriptions) and qualitative data thematic synthesis (analytical themes). Integration of data syntheses will inform discussion.

**Funding:** The author(s) received no specific funding for this work.

**Competing interests:** The authors have declared that no competing interests exist.

## Implications

This systematic review will provide insight into patient satisfaction with APP internationally, exploring attributes that influence satisfaction. This will aid design, implementation, or improvement of APP and facilitate the delivery of patient-centred, high-quality healthcare. Lastly, this review will inform future methodologically robust research investigating APP patient satisfaction and experience.

## Introduction

### Rationale

Patient satisfaction is a complex construct measuring how pleased or happy a patient feels about care they received, evaluated relative to their expectations and needs [1–3]. Patient satisfaction is defined by Hills and Sachs (2007) as "a sense of contentedness, achievement or fulfilment that results from meeting patients' needs and expectations with respect to specific and general aspects of health care" [4]. Patient satisfaction largely exists within subjective (epistemology) and relativist (ontology) paradigms, allowing patients to share how they felt about their care, perhaps distinct from the quality of care provided. Ng & Luk (2019) suggest patient satisfaction consists of human and system attributes. Human attributes may be attitudinal including practitioners' courtesy, friendliness, kindness, and approachability, or related to practitioners' technical competence demonstrated through professional knowledge or adherence to high standards. System attributes may relate to accessibility, hygiene, or comfort of healthcare services, or their efficacy in improving or maintaining a particular health status [5]. These attributes align with findings from a qualitative review by Rossettini et al. (2020) whereby human and professional competence, communication, partnership of care, organisation of care, and physical environment were all identified as influencing factors of patient satisfaction [6]. These patient satisfaction attributes principally map across to relational (human) and functional (system) dimensions of service quality in non-clinical settings. In this context, the relational dimension refers to "customers' emotional benefits, beyond the core performance, related to the social interaction of customers with employees" and the functional dimension refers to "efficiency with which the service is delivered" [7]. Patient satisfaction is an outcome measure of a patients experience of care [3], mostly captured using surveys or questionnaires, and sometimes incorporated into patient-reported outcome measures (PROMs) or patient-reported experience measures (PREMs) [8, 9].

Patient experience is a similar but separate construct defined by the Beryl Institute as "the sum of all interactions, shaped by an organisation's culture, that influence patient perceptions, across the continuum of care" [10] and is considered a more objective (epistemology), descriptive, process indicator reflective of the quality of care received [2, 3]. If patient satisfaction is a measure of what a patient feels should happen in a healthcare setting relative to their expectations, patient experience is a measure of what best-practice standards state should happen in that setting.

Across healthcare, advanced practice is defined as a higher level of practice demonstrating a high degree of autonomy and complex decision making, often underpinned by a post-registration master's level education or equivalent that encompasses the four pillars of clinical practice, leadership, education, and research [11]. Advanced physiotherapy practice (APP) first emerged in the 1970s and has since organically evolved into over 15 fields of practice, across 14 countries and territories worldwide [12, 13]. This diversity has led to variation in APP

scope, titles, competency, capabilities, and educational requirements. For example, although the educational route for advanced clinical practice is typically a post-graduate, masters level qualification, countries like the UK accept both formal-accredited and portfolio-based (equivalence) developmental routes [11]. World Physiotherapy, working officially with the World Health Organisation to represent and further the physiotherapy profession globally [14], recognise APP variation exists (e.g., aforementioned educational routes), but have identified common themes across clinical fields, countries, and contexts [15]. These themes describe APP as a higher level of practice, perhaps associated with certain occupational titles (Table 1), requiring advanced clinical and analytical skills that influence service improvement and provide clinical leadership, resulting in safe and competent delivery of care to patients with complex needs. World Physiotherapy also recognise APP involves cross-pillar working (e.g., research and education) with the potential for advanced physiotherapy practitioners to perform tasks traditionally completed by other healthcare professionals [15]. These tasks may involve making and communicating diagnoses, triaging for surgical opinion, ordering diagnostic imaging, or prescribing and administering of medications [16]. While accepting this inherent breadth and variation, Table 2 conceptualises APP for this systematic review using the Tidier checklist [17].

Research investigating patient satisfaction with APP to-date lacks in-depth analysis, is specific to individual clinical fields (e.g., MSK or emergency department), and is limited to systematic reviews of mostly non-randomized and descriptive studies [20–26]. In these systematic reviews, patient satisfaction of MSK APP is rated highly, or as either equal to or more satisfactory than care provided by medical or orthopaedic surgical practitioners [20, 21, 24, 25]. However, methodological quality varies greatly within the included primary studies, and patient satisfaction is mostly measured using non-standardised or non-validated, locally devised tools. A systematic review investigating adult spinal pain by Lafrance et al. (2021) showed patients were highly satisfied with care provided by advanced physiotherapy practitioners, with some preference for APP over usual medical care. However, included studies were mostly of moderate quality (high = 1; moderate = 14; low = 3) and heterogeneity existed within measurement tools used to capture patient satisfaction (e.g., Likert, VSQ-9, perceived quality of care questionnaire). A theme that emerged from this review was the amount of time advanced physiotherapy practitioners spent with patients. APP consultation times were consistently longer than medical or surgical consultations, perhaps enabling more education and advice to be provided to patients, and in-turn driving satisfaction. Similarly, a systematic review investigating MSK physiotherapy in emergency departments by Matifat et al. (2019)

**Table 1. Examples of titles used to describe APP [13].**

| Title |
| --- |
| Advanced physiotherapy practitioner |
| Consultant physiotherapist |
| Extended scope practitioner |
| Specialist physiotherapist[a] |
| Highly specialist physiotherapist[a] |
| Advanced (practice) physiotherapist |
| Manual therapist[a] |
| Military physician extender[a] |
| Senior physiotherapist fellow[a] |

[a] Less common titles used to describe advanced practice physiotherapy by member organisations of World Physiotherapy [13].

**Table 2. TIDieR checklist for APP as an intervention.**

| Item | Description |
|---|---|
| Brief name | Advanced physiotherapy practice (APP) |
| Why? | To help meet the global demand placed on healthcare systems from burden of disease, people living longer, people living with higher levels of disability, and people with complex, significant rehabilitation needs [18]. |
| What (materials)? | Context specific and may include; educational resources handed to patients, exercise equipment, therapeutic devices, medical equipment for the administration of medications, or pharmaceutical products. |
| What (procedures)? | Context specific and may include; examination/assessment[a], making and communicating diagnoses and prognoses[a], intervention/treatment[a], re-examination[a], referral for diagnostic imaging, listing for surgery, referral into secondary or tertiary care, prescribing and/or administering of medications, and discharge[a]. |
| Who provided? | Highly experienced, autonomous physical therapists with post-registration, master's level (or equivalent) specialisation in clinical practice, education, leadership and management, and research. |
| How? | Mostly in face-to-face clinical settings, with some APP's taking place via virtual or telephone-based services. |
| Where? | Context specific and may include; inpatient hospital, outpatient hospital/clinic, patient home, community clinic, primary care, secondary care, tertiary care, within public, private, or third healthcare sectors, and in countries or territories globally with recognised APP. |
| When and how much? | Context specific and may involve a single or multiple appointments depending on the service and patient needs. |
| Tailoring | APP is patient-centred and tailored to individuals based on the service and patient needs. |
| Modifications | N/A |
| How well (planned) | Context specific and pragmatic dependent on the service and patient needs. |
| How well (actual) | Context specific and pragmatic dependent on the service and patient needs. |

[a] Physiotherapy procedures that may be included in APP [19].

highlighted elements of the clinical encounter where higher levels of patient satisfaction were reported with APP over usual medical care. Although limited to five studies of weak to strong evidence, patients appeared to prefer first aid advice, receiving written information regarding injury, and discharge to be provided by an advanced physiotherapy practitioner, and medications management by a physician. Again, patients appreciated time to ask questions of an advanced physiotherapy practitioner regarding their condition, that may not have been reciprocated during physician led consultations. Interestingly, patient satisfaction did not differ between APP and physician led care at 2 or 6 week follow up. Lastly, a systematic review by Trøstrup et al. (2020) aligned with findings from other reviews, showing patients in an orthopaedic diagnostic setting were highly satisfied with APP assessment, with consistent results from high, moderate, and low risk of bias studies. From this scoping review of the literature, patient satisfaction with APP appears to be high, but is mostly conducted in MSK clinical fields, using quantitative methods, and studies lack methodological quality. Considering that patient satisfaction provides insight into patient experience, itself a key component of evaluating healthcare quality, and that no focussed mixed studies synthesis of patient satisfaction with APP across clinical fields exists, further investigation is warranted.

## Objectives

1. To evaluate patient satisfaction with APP internationally.

2. To evaluate human and system attributes of patient satisfaction with APP.

## Materials and methods

This protocol has been reported in line with the Preferred Reporting Items for Systematic review and Meta-Analysis Protocols (PRISMA-P) [27, 28] and is registered with the international prospective register of systematic reviews (PROSPERO: CRD42023443612).

### Researcher positionality

All authors hold physiotherapy qualifications and mostly work at Advanced or Consultant level of practice. The first author is a UK-based physiotherapist with 16 years of clinical experience, 12 of which specialising in musculoskeletal outpatients. They hold an MSc in Sport and Exercise Medicine and are currently studying towards a PhD in Physical Therapy. The author group are based in the UK and Canada, with strong links to the USA and Australia. Academic experience varies within the group from PhD student to professorship in Physical Therapy, with experience in systematic mixed studies review methodology.

### Design

A systematic mixed studies review will be conducted using a parallel-results convergent design as described by Hong et al. (2017). In this design, qualitative and quantitative patient satisfaction data will be analysed separately and brought together during the interpretation of results in the discussion. Part one will address the first objective and will comprise of quantitative data from surveys, questionnaires, PROMs, and PREMs. Part two will address the second objective and will comprise of qualitative data from textual descriptions and interviews. A systematic mixed studies review is appropriate for this study as mixed evidence allows comprehensive understanding of complex phenomena (patient satisfaction), and because the objectives are not measuring effectiveness [29].

### Eligibility criteria

The study eligibility criteria (Table 3) were informed by the PICOS framework [30].

### Information sources

The Medline, Embase, CINAHL, Cochrane, Web of Science, and PEDro databases will be searched from inception to 17/8/2023. Grey literature will be searched through ProQuest Dissertations and Theses and trial registers (clinicaltrials.gov and World Health Organization International Clinical Trials Registry Platform). Manual searching the reference lists of

**Table 3. Eligibility criteria.**

| | |
|---|---|
| Participants | Patients of APP services from countries, territories or jurisdictions with recognised APP roles who are able to give informed consent. Studies of parent or carer satisfaction of APP services on behalf of a dependent will be excluded. |
| Intervention | APP as described by World Physiotherapy [15] whereby practitioners a) have advanced clinical and analytical skills that influence service improvement and provide clinical leadership, b) have post-registration masters level specialisation (or equivalence), c) deliver safe, competent care to patients with complex needs and d) may be associated with particular occupational titles (Table 1), in all clinical fields (e.g., MSK, orthopaedics, emergency department). |
| Comparison | Not applicable. |
| Outcome | Patient satisfaction. Studies investigating only patient experience will be excluded. |
| Study designs | Primary research of any design (e.g., experimental, qualitative, or mixed-methods). |
| Publication language | English, or papers able to be sufficiently translated into English via Google Translate [31]. |

included studies will identify any further articles that meet eligibility criteria. Expert researchers of APP and/or patient satisfaction will be consulted for identification of other potential studies. If studies cannot be retrieved, access requests will be made through contacting the authors by email on two occasions.

## Search strategy

Database search strategies were developed first by the lead author, then reviewed and revised by an independent specialist librarian not otherwise associated with the systematic review [32]. Table 4 demonstrates the MEDLINE (Ovid) search strategy that will be adapted to meet search terms of other included databases (see S1 Table).

## Study records

**Data management.** Covidence (Covidence systematic review software, Veritas Health Innovation, Melbourne, Australia), a web-based collaboration software platform that streamlines processes within systematic and other literature reviews, will be used to import all citations, remove duplicates yielded from the search strategy, and aid in determining eligibility during screening and review stages.

**Selection process.** Pre-screening eligibility criteria training and instruction will be provided to reviewers before selection to promote consistency. CD will perform searches and import citations into Covidence. Two reviewers (CD and JM) will independently screen titles and abstracts against eligibility criteria. Disagreement on studies to be included will be resolved through discussion. A third author (TN) will provide arbitration for any unresolved discrepancies. Full text reports will be obtained for all studies deemed to meet eligibility criteria from title and abstract screening. If it is not possible to determine if a study meets eligibility criteria from title and abstract screening, they will be marked "maybe", full text reports will be obtained, and they will be included in the full text review. Full texts will then be independently

**Table 4. MEDLINE (Ovid) search strategy.**

| 1 | Patient Satisfaction/ |
|---|---|
| 2 | ((patient* or client* or participant*) adj5 (experience* or perception* or perceive* or expectation* or happiness or contented* or fulfilment)).tw,kf. |
| 3 | Satisf*.tw,kf |
| 4 | 1 or 2 or 3 |
| 5 | Physical Therapists/ |
| 6 | Physical Therapy Specialty/ |
| 7 | Physical Therapy Modalities/ |
| 8 | physiotherap*.tw,kf. |
| 9 | physio-therap*.tw,kf. |
| 10 | physical therap*.tw,kf. |
| 11 | 5 or 6 or 7 or 8 or 9 or 10 |
| 12 | 4 and 11 |
| 13 | ((advanced or speciali* or consultant* or extended or expanded) adj5 (practice or practitioner* or scope or role or physi*)).tw,kf. |
| 14 | military physician extender*.tw,kf. |
| 15 | senior physiotherapist fellow*.tw,kf. |
| 16 | (First adj3 contact adj3 (physiotherap* or physio-therap* practitioner*)).tw,kf |
| 17 | 13 or 14 or 15 or 16 |
| 18 | 12 and 17 |

reviewed by two reviewers (CD and JM) against eligibility criteria. Disagreement on full texts to be included will be resolved through discussion, or through third author (TN) arbitration. Any overlapping or companion studies (multiple publications for one study) will be accounted for in the final data synthesis. Inter-rater agreement will be calculated using Cohens kappa (k) at both the title and abstract screening and full text review stages. Study identification, screening, eligibility, and inclusion will be presented using a PRISMA flow diagram (Fig 1).

**Data collection process.** Two reviewers (CD and JM) will extract data independently and in duplicate from each eligible study. Data will be extracted into a standardised form with preset data items. Prior to full data extraction, a pilot of 5 studies will be completed by both reviewers to ensure consistency. Disagreement on data extraction will be resolved through discussion, or through third author (TN) arbitration. Authors will be contacted on two occasions to account for any missing or incomplete data (if necessary).

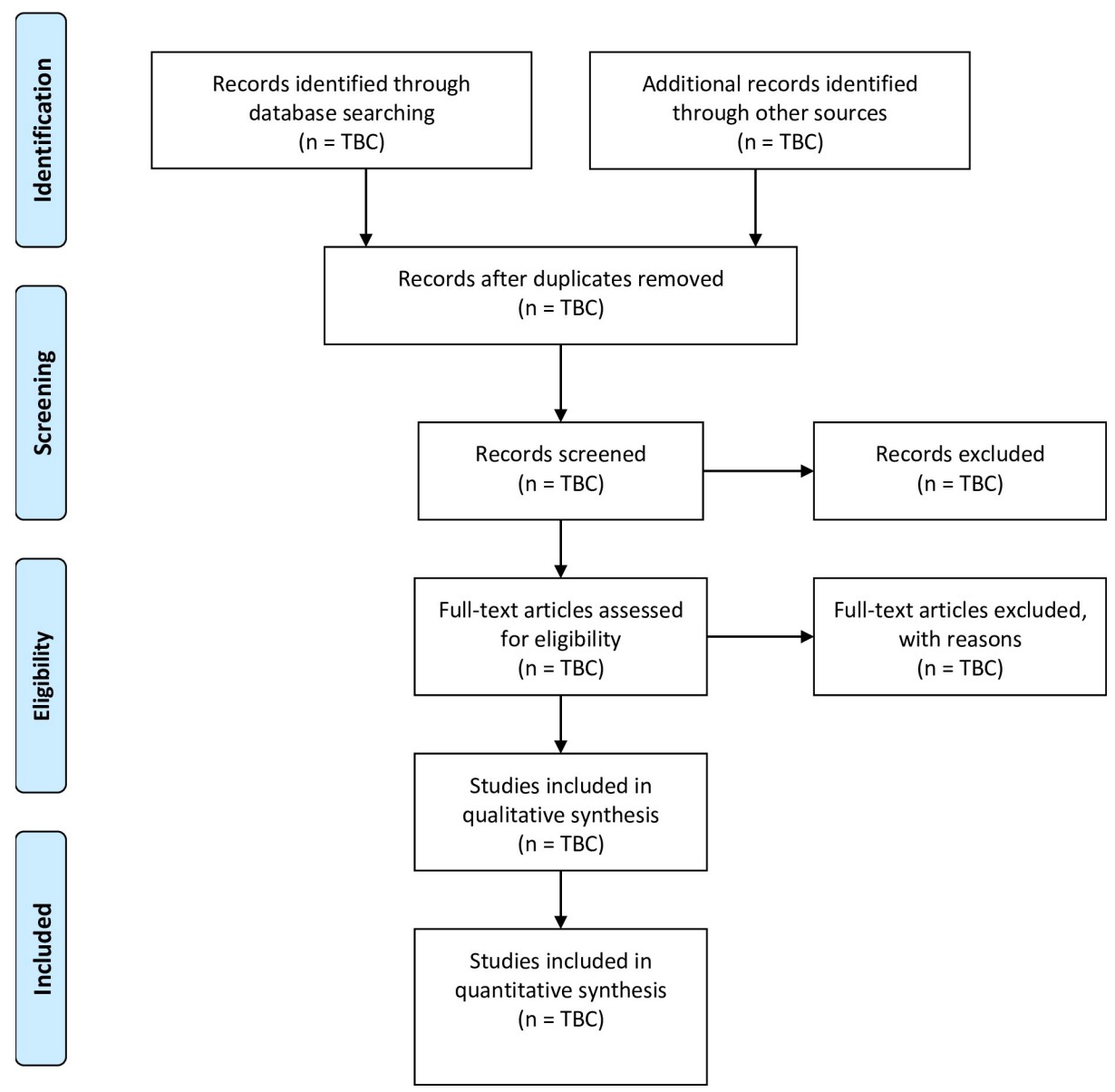

**Fig 1. PRISMA flow diagram [33].** TBC = To be confirmed.

**Table 5. Data items.**

| Data items |
| --- |
| Author(s) |
| Year of publication |
| Country of data collection |
| Study design |
| Study setting |
| Interaction type |
| Number of interactions with APP |
| APP consultation time |
| Patient characteristics |
| Eligibility criteria |
| Sample size |
| APP characteristics (incl. training) |
| Patient satisfaction data collection method and time-point |
| Patient satisfaction results |
| Human attributes of patient satisfaction |
| System attributes of patient satisfaction |

**Data items.**   Data items to be collected are listed in Table 5.

**Outcomes and prioritisation.**   The outcome used in this systematic review is patient satisfaction with APP measured using surveys, questionnaires, PROM's, or PREMS.

**Quality assessment.**   The mixed methods appraisal tool (MMAT) version 2018 will be used for methodological quality assessment of the included studies. The MMAT is a valid, reliable, and efficient tool [34, 35] developed to appraise the methodological quality of five empirical study categories (qualitative research, randomised controlled trials, non-randomised studies, quantitative descriptive studies, and mixed methods studies) eligible for inclusion [36]. Using a single critical appraisal tool for all included studies will allow a consistent approach to methodological quality assessment. Two reviewers (CD and JM) will become familiar with the MMAT before independently completing a methodological quality assessment of each included study. A pilot of the first 5 studies will be completed to ensure consistency between reviewers, and any discrepancies resolved through discussion. A detailed, descriptive presentation of ratings for each quality criterion will be provided to inform the overall strength of the evidence, rather than a numeric score. This descriptive approach, as recommended by the MMAT developers, will be used for two main reasons. Firstly, presenting a single score, or a summary scale made up of somewhat arbitrary thresholds, may hide significant inadequacies within the study and overstate the trustworthiness of the evidence [37]. Secondly, although quantifying quality criteria may be appropriate for qualitative research situated within positivist or post-positivist paradigms, it is less suitable for quality assessment of qualitative research existing in other paradigms [38]. Owing to the descriptive presentation of ratings, the constructivist nature of this research, and to ensure this review includes the totality of evidence available in the synthesis, studies will not be excluded based on methodological quality. Instead, quality assessment will be used to demonstrate transparency in the results, contribute to the robustness of the synthesis, and to nuance or weight conclusions and recommendations from this research [39].

**Data synthesis.**   The parallel-results convergent synthesis design involves the separate synthesis of quantitative and qualitative data, before integration in the discussion as detailed below [40].

**Quantitative synthesis.** To address the first objective a narrative synthesis as proposed by Popay et al. (2006) will be conducted. This synthesis method has been chosen due to the expected heterogeneity of study design, participant characteristics, setting, APP characteristics, and/or data collection method used in primary studies, including an expected lack of randomised clinical trials. The methodological foundation of this narrative synthesis will include 3 key elements, completed iteratively and non-sequentially [41].

*Developing a synthesis of findings from included studies.* This element will initially describe the relevant patient satisfaction data from included studies using statistical data and textual descriptions of findings. Studies will be organised in a way that allows patterns in the data to emerge (e.g., by clinical field, country, or study design). Where studies present patient satisfaction data using homogenous rating scales like the VSQ-9 [42], data within groups may be pooled.

*Exploring relationships within and between studies.* Patterns that emerge from the data will then be interrogated, exploring factors that may influence patient satisfaction with APP. This will involve looking for relationships between study characteristics and the patient satisfaction results, then comparing and contrasting these relationships across different studies. The influence of heterogeneity will be explored, considering variations in study design, participant characteristics, setting, APP characteristics, data collection method, and other context relating to patient satisfaction.

*Assessing the robustness of the synthesis.* MMAT quality criterion outcomes, alongside the quantity of studies included, will be used to provide a summary of the robustness of the synthesis. Methodological quality will be built into the exploration of heterogeneity, and if appropriate, higher methodological studies will be given larger weighting in the final interpretation of results.

**Qualitative synthesis.** To address the second objective a thematic synthesis as proposed by Thomas and Harden (2008) will be conducted. This synthesis method has been chosen to integrate the findings of multiple qualitative studies or qualitative data from multiple mixed methods studies and will involve three key steps [43].

*Free line-by-line coding.* Free line-by-line coding will take place using all text within the "results" or "findings" sections in qualitative studies of patient satisfaction (including qualitative sections of mixed-methods studies).

*Construct descriptive themes.* Free codes will be organised and grouped into related areas to construct descriptive themes. During free line-by-line coding and constructing descriptive themes, the synthesis will stay close to the data, be somewhat descriptive, and use in-vivo coding where possible.

*Development of analytical themes.* To synthesise the data beyond the content of the original studies, a constructivist paradigmatic approach will be used to develop analytical themes. Constructivism assumes multiple, equally valid realities exist and that meaning is hidden within the data and must be discovered through reflection and interpretation [44]. This constructivist approach will allow a deeper synthesis of the data that can answer the second objective of this systematic missed studies review. Two reviewers (CD and JM) will complete this thematic synthesis independently, before bringing analytical themes together to discuss, debate, and explore similarities and differences. The final stage of the synthesis using crystallisation will develop and create more abstract, analytical themes that have substance and offer rich accounts of patient satisfaction attributes [45]. To increase integrity and align with constructivist approaches, both reviewers will use reflexivity to situate themselves in the research and evaluate how subjective and intersubjective elements may influence the shaping of analytical themes [46]. MMAT criteria will be presented, allowing readers to interpret synthesis findings and judge trustworthiness in their own contexts. The relative contribution of qualitative studies with low methodological quality to final analytic themes will also be presented.

**Meta-biases.**   Two important meta-biases relevant to this systematic review are selection and publication biases. Selection bias will be minimised by conducting a broad, inclusive, and exhaustive search from published and grey literature sources, and by using multiple reviewers during the selection process. Publication bias will be reviewed by comparing patient satisfaction with APP between published and any un-published studies identified in grey literature, where homogenous data are available [27].

## Discussion

Integration of quantitative and qualitative findings will occur during the discussion. Triangulation "as a methodological metaphor", whereby points of the triangle are represented by theoretical concepts and empirical data from qualitative and quantitative syntheses, will be used to identify relationships between findings and theory, and provide deeper insight into patient satisfaction with APP internationally [47].

Depth of understanding of patient satisfaction will allow greater consideration of how pleased or happy patients are with APP care they receive, across clinical fields. It will also provide insight into attributes that may influence patient satisfaction with APP. Knowledge of these attributes should improve our understanding of patient's expectations and needs when interreacting with APP. This improved awareness will aid design, implementation, or improvement of APP and facilitate the delivery of patient-centred healthcare. Being more sensitive to patient satisfaction and patient experience in APP will also drive the quality of healthcare being delivered. Lastly, this systematic review will inform future methodologically robust research investigating APP patient satisfaction and experience.

## Supporting information

**S1 Table. Adapted search strategies.**
(DOCX)

**S2 Table. Completed PRISMA-P checklist.**
(DOCX)

## Acknowledgments

Alanna Marson, Research & Scholarly Communication Librarian at Western University (London, Ontario, Canada) for support with reviewing and shaping the search strategy.

## Author Contributions

**Conceptualization:** Chris Davis, Tim Noblet, Jai Mistry, Katie Kowalski, Alison Rushton.

**Methodology:** Chris Davis, Tim Noblet, Jai Mistry, Katie Kowalski, Alison Rushton.

**Supervision:** Tim Noblet, Katie Kowalski, Alison Rushton.

**Writing – original draft:** Chris Davis, Alison Rushton.

**Writing – review & editing:** Chris Davis, Tim Noblet, Jai Mistry, Katie Kowalski, Alison Rushton.

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
