## [Decision Letter · Decision Letter 0]

4 Sep 2023

PONE-D-23-21887Patient satisfaction with advanced practice physiotherapy internationally: protocol for a systematic mixed studies reviewPLOS ONE

Dear Dr. Davis,

Thank you for submitting your manuscript to PLOS ONE. After careful consideration, we feel that it has merit but does not fully meet PLOS ONE’s publication criteria as it currently stands. Therefore, we invite you to submit a revised version of the manuscript that addresses the points raised during the review process.

We look forward to receiving your revised manuscript.

Kind regards,

Maher Abdelraheim Titi

Academic Editor

PLOS ONE

Journal Requirements:

Reviewers' comments:

Reviewer's Responses to Questions

**Comments to the Author**

1. Does the manuscript provide a valid rationale for the proposed study, with clearly identified and justified research questions?

Reviewer #1: Yes

Reviewer #2: Yes

2. Is the protocol technically sound and planned in a manner that will lead to a meaningful outcome and allow testing the stated hypotheses?

Reviewer #1: Yes

Reviewer #2: Yes

3. Is the methodology feasible and described in sufficient detail to allow the work to be replicable?

Reviewer #1: Yes

Reviewer #2: Yes

4. Have the authors described where all data underlying the findings will be made available when the study is complete?

Reviewer #1: Yes

Reviewer #2: Yes

5. Is the manuscript presented in an intelligible fashion and written in standard English?

Reviewer #1: Yes

Reviewer #2: Yes

6. Review Comments to the Author

You may also provide optional suggestions and comments to authors that they might find helpful in planning their study.

Reviewer #1: Dear Authors,

Thanks a lot for the opportunity you have offered me to revise the fascinating protocol "Patient satisfaction with advanced physiotherapy practice internationally: protocol for a systematic mixed studies review". I thank the authors for their efforts in producing this exciting protocol. It is perfectly aligned with my area of research and expertise; thus, I am confident I can offer a valuable peer review.

As a significant strength, this manuscript aims to evaluate patient satisfaction with advanced physiotherapy practice (APP) internationally and to evaluate human and system attributes of patient satisfaction with APP. This proposal is interesting in the field and adds information to the existing evidence in the literature.

As a major weakness, the manuscript sometimes lacks details and clarity concerning methodological steps that would help improve the understanding of the manuscript. Therefore, I have suggested some strategies to improve authors' reporting and increase the quality of their work.

Overall, my peer review is a minor revision: I suggest revising the manuscript to improve the pitfalls presented. The final goal is to improve the overall clarity of the message to help the reader understand this fundamental topic. I look forward to reading the revised version of the manuscript.

Thanks again, and good luck with researching in this challenging time.

¶MINOR REVISION

#INTRODUCTION

*Background: The authors analyse the concepts of patient satisfaction and patient experience in the context of physiotherapy. Although the work they do is commendable, I believe it is appropriate to integrate some fundamental references for both concepts into the background. Among them, I suggest: patient satisfaction (doi: 10.1080/09638288.2018.1501102), patient experience (doi: 10.1093/ptj/pzac080 doi: 10.1186/s40945-020-00088-6).

#METHODS

*Tidier: Please report it in full.

*Search strategy: Please report the search strategy for each database.

*Research group details: Please report details about the authors' background. Are they pt? Have they experienced in performing mixed-method systematic reviews?

#FIGURE

*Figure 1: Please report in full the abbreviation TBC.

Reviewer #2: This is an interesting protocol on patient satisfaction with advanced practice physiotherapy. I have one major comment that would need to be addressed before publication; otherwise, I provide 12 other comments to improve your protocol and eventual paper but that can be considered as minor comments/suggestions.

Major comment

1. In your eligibility criteria you include “APP (…) have post-registration masters level specialization”, however, many studies include APP without this specialization and who instead has a bachelor level education with an onsite APP training such as a residency type training. Other studies poorly described the training of APP, sometimes because many APP with different training are included. Therefore, this inclusion criterion, if taken by its words, would lead to significant number of relevant papers to be excluded from your analysis. It is also important to make the distinction between advanced practice physiotherapy roles, which does not necessarily require a specific training or highly specialized training and advanced practice physiotherapy title, that are for example not present in Canada.

Minor comments / suggestions:

1. Lines 29-30 “However, no evidence synthesis of patient satisfaction with APP across clinical fields exists.” This is no completely true as a few systematic review included a synthesis of patient satisfaction, at least of quantitative questionnaires. You also discussed results from a few studies in the introduction.

2. Lines 39-40 & 85-86 “b) have post-registration masters level specialisation” Does that mean that you will exclude all studies including advanced practice physiotherapists that do not have a masters level specialisation? If so, you will exclude some relevant studies as many include physiotherapists with a bachelor level education who received a non-master-level APP training (e.g., residency type training). Please consider removing this eligibility criteria or clarifying the text.

3. Lines 44-45: Please provide more details regarding the data synthesis plan.

4. Line 103 – Table 1: Although they were reported in the Tawiah et al. paper, are ‘Manual Therapists’, ‘Military Physician Extender’ and ‘Senior Physiotherapist Fellow’ really terms used to describe APP? I would suggest either to remove them from Table 1 or to provide a footnote providing more details (e.g., these terms have been used in the past to describe APP although …)

5. Line 105 – Table 2: in the What (procedures), we see treatment and rehabilitation that are under the scope of PT and may or may not be included in an APP model of care (although they are often included). Should a distinction be made between APP procedures (like the other ones that are described) and PT procedures that may be included in an APP model of care?

6. Line 105 – Table 2: As mentioned above, it is not necessary to be an “highly experienced, physical therapists with post-registration master’s level (or higher)” to be considered as an advanced practice physiotherapist or at least to work in an advanced practice role.

7. Lines 112-113: You discussed well the results from previous systematic reviews. However, there is two limitations in the previous systematic review that I think you should highlight. First, none of them conducted a comprehensive evaluation of patient satisfaction with APP care, instead they all focused on various outcomes specific to a clinical setting (specialized medical care or emergency department) which limit the evaluation of satisfaction to these settings. Also, they only included quantitative data, omitting to include qualitative analysis from patients interviews, for example. I think you can also cut a bit the description of these studies, especially if you are lacking word space.

8. Line 159 – Table 3: Intervention same comment as above regarding the masters level specialisation

9. Line 159 – Table 3: Study design, you could add a few examples e.g., such as experimental, observational or qualitative studies

10. Line 159 – Table 3: Which software? Also in the final article, it would be important to provide details regarding the studies that were translated and included or excluded because translation was not possible.

11. Lines 172-175: You provide the search strategy for MEDLINE and mentioned that the search strategy was adapted for other included databases. Although it is not of common practice, it would be even more transparent and useful to other researchers who might want to replicate your study to provide all search strategies in supplementary materials (at least for the paper publication).

12. Lines 280-282: You mention that a “constructivist paradigmatic approach” will be used. Although you described why you would use it, I believe that readers would benefit from having a short description of what is a “constructivist paradigmatic approach”. Also, the reference 40 (Thomas & Harden 2008) does not describe this approach.

7. PLOS authors have the option to publish the peer review history of their article (what does this mean?). If published, this will include your full peer review and any attached files.

Reviewer #1: No

Reviewer #2: **Yes: **Simon Lafrance

---

## [Author Response · Author response to Decision Letter 0]

3 Oct 2023

Please see attached rebuttal letter titled "response to reviewers" for our responses to all specific reviewer and editor comments.

---

## [Decision Letter · Decision Letter 1]

9 Oct 2023

Patient satisfaction with advanced physiotherapy practice internationally: protocol for a systematic mixed studies review

PONE-D-23-21887R1

Dear Dr. Davis,

We’re pleased to inform you that your manuscript has been judged scientifically suitable for publication and will be formally accepted for publication once it meets all outstanding technical requirements.

Kind regards,

Maher Abdelraheim Titi

Academic Editor

PLOS ONE

Reviewers' comments:

Reviewer's Responses to Questions

**Comments to the Author**

1. Does the manuscript provide a valid rationale for the proposed study, with clearly identified and justified research questions?

Reviewer #1: Yes

Reviewer #2: Yes

2. Is the protocol technically sound and planned in a manner that will lead to a meaningful outcome and allow testing the stated hypotheses?

Reviewer #1: Yes

Reviewer #2: Yes

3. Is the methodology feasible and described in sufficient detail to allow the work to be replicable?

Reviewer #1: Yes

Reviewer #2: Yes

4. Have the authors described where all data underlying the findings will be made available when the study is complete?

Reviewer #1: Yes

Reviewer #2: Yes

5. Is the manuscript presented in an intelligible fashion and written in standard English?

Reviewer #1: Yes

Reviewer #2: Yes

6. Review Comments to the Author

You may also provide optional suggestions and comments to authors that they might find helpful in planning their study.

Reviewer #1: Dear Authors,

congratulations. The paper is suitable for publication.

You have done a good job with your revision.

Best regards.

Reviewer #2: The author adequately address my comments. The description of APP in the introduction (master or equivalency) is adequate, but I believe that these details should also be present in the method section (e.g., what do you consider as an equivalence, this could be added as a footnote). A reader that reads only the method section should be able to know clearly what is included in your review. Although I believe this should be clarified, I do not think the paper should go in another revision. The authors may decide or not to make corrections.

I think there is a mistake at line 166: "They" should be "He" ?

7. PLOS authors have the option to publish the peer review history of their article (what does this mean?). If published, this will include your full peer review and any attached files.

Reviewer #1: No

Reviewer #2: **Yes: **Simon Lafrance

---

## [Editor Report · Acceptance letter]

12 Oct 2023

PONE-D-23-21887R1 

Patient satisfaction with advanced physiotherapy practice internationally:  protocol for a systematic mixed studies review 

Dear Dr. Davis:

I'm pleased to inform you that your manuscript has been deemed suitable for publication in PLOS ONE. Congratulations! Your manuscript is now with our production department. 

Kind regards, 

on behalf of

Dr. Maher Abdelraheim Titi 

Academic Editor

PLOS ONE